# EWOD Chip with Micro-Barrier Electrode for Simultaneous Enhanced Mixing during Transportation

**DOI:** 10.3390/s23167102

**Published:** 2023-08-11

**Authors:** Shang Gao, Xichuan Rui, Xiangyu Zeng, Jia Zhou

**Affiliations:** 1School of Microelectronics, Fudan University, Shanghai 200433, China; sgao21@m.fudan.edu.cn (S.G.); 18112020038@fudan.edu.cn (X.R.); 2Department of Micro/Nano Electronics State Key Laboratory of Radio Frequency Heterogeneous Integration, Shanghai Jiao Tong University, Shanghai 200433, China

**Keywords:** micro-barrier, mixing, EWOD, microfluidic

## Abstract

Digital microfluidic platforms have been extensively studied in biology. However, achieving efficient mixing of macromolecules in microscale, low Reynolds number fluids remains a major challenge. To address this challenge, this study presents a novel design solution based on dielectric electro-wetting (EWOD) by optimizing the geometry of the transport electrode. The new design integrates micro-barriers on the electrodes to generate vortex currents that promote mixing during droplet transport. This design solution requires only two activation signals, minimizing the number of pins required. The mixing performance of the new design was evaluated by analyzing the degree of mixing inside the droplet and quantifying the motion of the internal particles. In addition, the rapid mixing capability of the new platform was demonstrated by successfully mixing the sorbitol solution with the detection solution and detecting the resulting reaction products. The experimental results show that the transfer electrode with a micro-barrier enables rapid mixing of liquids with a six-fold increase in mixing efficiency, making it ideal for the development of EWOD devices.

## 1. Introduction

In recent years, there has been extensive research on digital microfluidic platforms in the field of biology [1,2,3,4]. While these platforms are typically used in multiple steps for biological applications, the amount of time dedicated to mass transfer (i.e., enhanced mixing processes) plays a crucial role in determining sensor efficiency [5,6]. Mixing is commonly necessary in chemical and biological reactions for sample dilution and reagent homogenization. It plays a key role in increasing the efficiency of polymerase chain reaction (PCR), DNA hybridization analysis, and drug delivery, as well as promoting the sensitivity and induction speed of detections. Although pure diffusion mass transfer is sometimes employed in microfluidic devices, it is a slow process that requires long channels to achieve satisfactory mixing. At microscopic scales, fluids have low Reynolds numbers (*R*e) and are typically highly ordered and laminar, meaning that viscous forces greatly exceed inertial forces, thereby suppressing any irregular flows that could induce fluid mixing. While diffusion mixing of small molecules (and therefore rapidly diffusing species) can occur within seconds over tens of micrometers, the mixing of larger molecules such as peptides, proteins, and high-molecular-weight nucleic acids may require several minutes to hours of equilibrium time at considerable distances [7,8,9]. Moreover, the size of microfluidic devices prohibits them from taking advantage of the turbulent mixing observed in macroscopic systems. This urgent need for efficient mixing techniques in microfluidic devices has led to developments that allow shorter channels, independence from channels or other devices, and a more efficient lab-on-a-chip.

Currently, there have been numerous reports on the study of rapid mixing in low *R*e fluids. According to the different methods of mixing, microfluidic mixers can be divided into two types: passive mixing and active mixing. For passive mixing digital microfluidic platforms that require additional auxiliary platforms, pumps [10,11], valves [12], and other devices are typically required to implement the hybrid function. One of the simplest methods is to insert an obstacle into the flow channel, the addition of which causes the liquid in the flow channel to create vortices around the barriers [13]. The multiple fluids in the channel thus gain more contact area, promoting mixing. While this solution satisfies the requirement for fast mixing, the flow channel, obstacle array, and pump insertion are impossible tasks for the lab-on-a-chip. Not only that, such on-chip high area ratio duct arrays [14] or chamber arrays [15] are a great challenge for high integration on the chip. Digital microfluidic platforms with active mixing usually use surface acoustic waves (SAW) [16,17], photoelectric wetting [18,19], thermal capillary forces [20,21], magnetic beads [22,23,24] and other methods when dealing with droplets. Although these methods offer the possibility of complete operation on a single chip, they still have many problems, such as high operating voltage, insufficient control accuracy, potential contamination, and high operating temperatures that may be harmful to biological survival.

However, electrical wetting on dielectric (EWOD) has the potential to revolutionize digital microfluidic platforms due to its fast response, simplicity, and ultra-high accuracy. Currently, EWOD-based digital microfluidics has a wide range of applications [25,26,27,28]. One approach to addressing the challenge of achieving fast mixing at low *R*e is by transporting the droplet on the electrode in large quantities and over long distances, which promotes good mixing within the droplet [29,30]. Nonetheless, this method suffers from excessive time consumption. Alternatively, combining EWOD technology with other microfluidics on the chip [31,32] can solve the mixing speed problem, but it requires additional operations or chip area, which poses a challenge for chip integration. Therefore, finding a solution to achieve fast mixing of low *R*e fluids using EWOD technology without additional assistance is an urgent issue that needs to be addressed. 

In this study, we proposed a new design scheme starting with the geometric design of the transport electrode to meet the need for rapid mixing of low *R*e fluids on an EWOD chip. A micro-barrier embedded in the transport EWOD electrode is the key to achieving the simultaneous enhanced mixing function during droplet driving, which provides higher integration for the EWOD chip. The study first verified the general mixing performance of the micro-barrier through dye mixing experiments. Then, the mixing capability of the micro-barrier was quantitatively analyzed by characterizing the motion state of the particles within the droplet. Finally, the new design was applied to the mixing of sorbitol and its detection reagents. The detection of the generated products was successfully performed. This design provides a new and bright solution for EWOD devices to achieve versatility and integration for biological applications.

## 2. Design and Fabrication

The Lippmann-Young equation [33,34] describes the relationship between the contact angle (*θ*) of droplets and the applied voltage in EWOD devices:cos*θ* − cos*θ*_0_ = *CV*^2^/2*γ*_LG_(1)
where *θ*_0_ is the contact angle without applied voltage, *C* is the surface capacitance between the electrode and the liquid (i.e., the specific capacitance of the dielectric layer that separates the electrode from the liquid), *γ*_LG_ is the interfacial tension between the liquid and the surrounding medium (e.g., air or oil), and *V* is the applied voltage.

According to the contact line (CL) theory and the principle of electric capillary force, the EWOD force [35,36,37] is:*F*_EWOD_ = *L*_E_ *γ*_LG_ (cos*θ* − cos*θ*_0_)(2)
where *L*_E_ is the length of CL.

Therefore, we proposed an asymmetric electrode design scheme with a micro-barrier, as shown in Figure 1a. The front end of the electrode extends the length of the CL with the droplet, enhancing the driving force on the droplet. The wedge-shaped part in the middle and lower parts is the micro-barrier, which forces the droplet to be squeezed and deformed during the backward movement, forming a vortex flow inside the droplet and enhancing the mixing ability of the electrode. The electrode has a concave shape at the posterior end for the next electrode to be embedded, forming an electrode array. The geometric contour of the electrode follows our previous research results and is designed to be an asymmetric shape [35], which keeps the projection CL on the active electrode at the front of the droplet longer than that at the rear, producing a non-vanishing electrocapillary force to activate the droplet to the right side of Figure 1a. Through careful design of the electrode geometry, the mixing ability of the electrode is enhanced without requiring additional area or repeated transportation. On the other hand, thanks to the asymmetric contour, long-distance transportation is achieved with only two opposing electrical signals (alternating grounding and activation). This further optimizes the use of the area on the chip and improves integration.

A simulation was conducted to verify the feasibility of this design, using COMSOL Multiphysics 5.4, to investigate the impact of a micro-barrier on the motion trajectory and the velocity direction of the internal flow field of a droplet.

The analysis showed that the motion of droplets during transportation comprises three main steps. Step 1 is when the droplet is forced to move horizontally along the CL direction under the influence of *F*_EWOD_. At this step, the velocity direction of the internal flow field of the droplet is primarily horizontal (Figure 2a), similar to ordinary rectangular electrodes. Step 2 is critical for achieving mixing effects, where the droplet is transported to a micro-barrier. Unlike the first step, the micro-barrier provides new *F*_EWOD_ in the vertical direction, resulting in an internal vortex flow and promoting mixing effects (Figure 2b). Once the droplets inevitably pass through the micro-barrier, they are transported across the electrode, generating new vortices. Here, we define the transmission of a droplet through an electrode (either a barrier or a rectangular electrode) as one-time mixing. In step 3, the droplet is transported to the tail end of the electrode, providing an opportunity for the droplet to make contact with the next electrode. Therefore, when the next electrode is activated, the droplet begins a new cycle of simultaneous mixing during its EWOD transportation.

For comparison, we analyzed the internal flow field of the droplet during the complete transfer process using conventional rectangular electrodes (see Appendix A). In the process of transporting droplets with rectangular electrodes. The velocity direction of the flow field inside the droplet is almost horizontal and cannot break the laminar flow to produce turbulence, resulting in poor mixing.

We fabricated a double plate EWOD Chip with a micro-barrier (Barrier EWOD Chip), illustrated in Figure 1b, using glass substrates coated with indium-tin-oxide for both the upper and lower plates. The electrode array on the lower plate was patterned by standard photolithography and wet etching methods. The dielectric and hydrophobic layers on the lower plate were created through spin coating methods, using SU8-2002 and Teflon materials, respectively. Assemble the upper and lower plates using a plug gauge to ensure a clearance height of 100 μm. The detailed manufacturing process is in the Appendix A. The Barrier EWOD Chip is controlled using a smartphone platform and is activated using AC power at a frequency of 1 kHz with a signal switching time of 50 ms. The EWOD chip with rectangular electrodes (Re EWOD Chip) was fabricated in the same way as the control group experiment. The size of the rectangular electrode is 2000 × 2000 μm.

## 3. Results and Discussion

### 3.1. General Mixing Capability of the Barrier EWOD Chip

The general mixing ability of the Barrier EWOD Chip was demonstrated by analyzing the degree of mixing of the blue dye in clear deionized water (DI water) and yellow DI water. As a control group experiment, the degree of mixing of the Re EWOD Chip was also recorded. The activation voltage used for all experimental groups was 240 Vpp. The droplet motion and internal color changes were recorded by a high-speed camera, as shown in Figure 3a,b.

Use ImageJ to process the captured image into grayscale, then analyze its grayscale dataset to calculate the standard deviation *φ* of the pixel grayscale values. The mixing degree is represented in *μ*_m_ (0 ≤ *μ*_m_ ≤ 1), and the calculation of *μ*_m_ is:*μ*_m_ = (*φ_i_* − φ*_s_*)/(*φ_f_* − *φ_s_*)(3)
where *i* represents the *i* times of mixing, *s* represents initial status, and *f* represents complete mixing. Complete mixing refers to multiple fully mixed blends. In this study, *μ*_m_ ≥ 0.99 was considered to be well mixed. 

Figure 4 demonstrates the variation in *μ*_m_ within the droplet with the number of times it mixes. The introduction of the micro-barrier significantly enhances the mixing within the droplet by altering the flow direction within the fluid at the micro-barrier, disrupting the laminar flow. This observation is consistent with the simulation results. The Barrier EWOD Chip achieves a good mixing state (*μ*_m_ = 0.996) within five times of mixing, taking 5.26 s. While, the Re EWOD Chip needs 30 times of mixing to achieve *μ*_m_ = 0.994, taking 42.89 s. The mixing efficiency of the Barrier EWOD Chip is six times that of the Re EWOD Chip. To rule out the influence of spontaneous diffusion, we analyzed the spontaneous diffusion of the dye within the droplet. It was observed that the diffusion velocity of the dye within the droplet was very low and had a negligible impact on the experiment. Even after 720 s, the blue dye had not completely diffused in 0.65 μL of DI water, with the *μ*_m_ being 0.68.

### 3.2. Effect of Barriers on Mixing

The effect of the barrier on mixing performance is described by adding fluorescent polystyrene particles to DI water and analyzing the impact of different *H*_b_ (barrier key sizes) on the rotation angle Δ*α* of the particles. The Δ*α* refers to the angle at which the particle rotates around the center of the droplet under the effect of the internal flow field of the droplet, from the initial position to the endpoint. Δ*α* is defined as an indicator of mixing performance. The transparency of all components allows for easy recording of particle locations. During the experiment, the Barrier EWOD Chip was placed under a fluorescent microscope to record video of the droplet and particle positions before and after each mixing. Then, Δ*α* is calculated using Kinovea. Three different particle sizes, 40, 60, and 80 μm in diameter, were used in the experiments, and the particles were dispersed at a low concentration into DI water before the experiments. A total of 0.65 μL of particle suspension was taken for each experiment, and before starting the experiment, the droplets were confirmed to contain only one particle under a fluorescent microscope.

Analysis of the experimental data revealed that there is a trend towards a bimodal distribution of Δ*α* for each particle size under different *H*_b_. Numerous experiments have shown that the inhomogeneity of the flow field inside the droplet leads to this distribution pattern. According to the experiments, the dividing line of the internal flow field is roughly in the 45° direction (Figure 5a), and the particles rotate counterclockwise around the center of the droplet (Figure 5b). After analyzing the experimental data, the region above the dividing line is defined as the low-speed region and vice versa as the high-speed region. The low-speed region means that when the initial position of the particle is in this region, it will obtain a lower initial speed during the mixing process, and vice versa. To eliminate the impact of the initial particle position on Δ*α*, the experiments were conducted by separately calculating the Δ*α* of the two regions within each experiment group and repeating the experiments three times for each group before calculating the mean value.

Figure 6 illustrates the impact of *H*_b_ on Δ*α*, revealing consistent trends across different particle sizes and regions. As *H*_b_ increases, Δ*α* exhibits a significant increase in both the low- and high-speed regions for each particle size. The maximum value of Δ*α* is observed at *H*_b_ = 600 μm. This enhancement in mixing performance is reflected in the increased *LE* in the vertical direction. According to Equation (2), LE is directly proportional to *F*_EWOD_, which leads to greater droplet deformation at the micro-barrier and an increase in *F*_EWOD_ in the vertical direction. Consequently, a stronger vortex field is generated within the droplet, allowing for increased contact area between liquids and facilitating better mixing. When *H*_b_ is 600 μm, considering the initial position of the particles, it is observed that the particles need to rotate around the center for a maximum of three times after mixing (i.e., the particles are located at the boundary between the low and high speed regions).

However, when *H*_b_ reaches 800 μm, Δ*α* starts to decrease. Analysis of the captured images reveals that this is due to an insufficient wetting area on the micro-barrier, preventing droplets from passing through and causing some droplets to overflow beyond the wetted area of the electrode. As a result, droplet deformation is reduced. In a control group experiment, Δ*α* was simultaneously recorded on the Re EWOD chip. The particles within the electrode primarily exhibited translational motion during droplet mixing, aligning with the direction of droplet motion. This translational motion confirms the presence of laminar flow and is consistent with the clear boundary observed during dye mixing.

It was also observed during the experiment that the activation voltage threshold increases with increasing *H*_b_, which is attributed to the decrease in wetted area.

### 3.3. Effect of Barriers on Activation Velocity

Mixing occurs as the droplet travels across the electrode, so the velocity at which the droplet crosses the electrode has a critical effect on the mixing velocity of the droplet. This study records the activation velocity of the droplet with different sizes of incorporated particles under different *H*_b_, with particle sizes of 40, 60, and 80 μm and a droplet volume of 0.65 μL. The activation velocity of the droplet is the ratio of the distance of the droplet from the inactive electrode to the next active electrode to the time, and the time consumed by the signal switching is not included in the total time. Each experiment was conducted using the threshold voltage (in Figure 6), and the droplet’s displacement and displacement time were recorded by a high-speed camera. The experiments were repeated seven times, and the mean value was calculated. 

The experimental results in Figure 7 show that the activation velocity of the droplet increases with increasing *H*_b_. This can be attributed to the fact that larger *H*_b_ values require a higher threshold voltage. Higher voltages result in greater *F*_EWOD_ and greater activation velocity. Further insights into the relationship between voltage and activation velocity can be found in the Appendix A. Interestingly, the presence of multiple particle sizes within the droplet has a minimal impact on the activation velocity of the droplet. To achieve higher activation velocities or efficiencies, extending the electrode array has been found to be effective. It has been observed that multiple droplets can be simultaneously activated and mixed on the electrode array when the spacing between droplets is at least one electrode. Please refer to Appendix A for related movies. It is worth noting that the volume of individual droplets that can be activated is limited to a range of 0.5 μL to 0.85 μL.

In addition, a change in the direction of droplet motion can be achieved by changing the electrode array orientation. This change comes from the electrode orientation, does not change the shape of the electrode and does not affect the mixing efficiency of the droplets.

### 3.4. Application of Mixing and Detection of Sorbitol

Sorbitol is widely present in animals, plants, microorganisms, and cultured cells and is not only one of the forms of sugar transport but also relevant to sorbitol resistance and food flavor [38,39]. Therefore, it is often necessary to detect changes in the content of sorbic acid in studies on sugar metabolism, biotic resistance, and food research.

The Barrier EWOD Chip was used for the rapid mixing of the sorbitol solution and detection solution, and then the ability of the chip to mix was verified by detecting the amount of generated product. The sorbitol concentration detection reagent kit used in the experiment was purchased from Beijing Solarbio Science & Technology Co., Ltd. (Beijing, China). Before the experiment, the sorbitol powder was dissolved in DI water to prepare a 2 mg/mL standard solution for use, and 2 mL of DI water was reserved for use in the waiting-to-be-tested tube (WT tube). During the experiment, 0.5 μL of the sorbitol solution was first dripped onto the chip, and then 0.15 μL of detection reagents 1 and 2 were added in sequence to form a mixture. After mixing the mixture 0 times, 5 times, 15 times, and 30 times using the Barrier EWOD Chip, the mixed droplets were respectively added to the WT tubes. As a control group experiment, equal amounts of sorbitol solution and assay reagent were mixed 30 times using the Re EWOD Chip and then added to the WT tube. Then, the WT tube was tested using a spectrophotometer (V650 from JASCO Corporation, Tokyo, Japan).

As depicted in Figure 8, it is evident that increasing the number of times you mix provides more opportunities for sorbitol to react with its detection solvent, which is particularly crucial for small droplets. When the number of times mixing reaches 30, the reactant undergoes a complete and full reaction, which can be confirmed by extra ultrasonic mixing. After 30 times of mixing, the solution tube was put in the ultrasonic sink for 3 min extra mixing. It was found that no additional chemical reaction occurred in the test tube (with an error of less than 1%, as shown in Figure 8). Therefore, it was confirmed that after 30 times of mixing, complete mixing had occurred in the droplets, with a mixing time of approximately 36.23 s. However, the yields after 30 times of mixing using the Re EWOD Chip were only comparable to those achieved with the Barrier EWOD Chip after five times of mixing, demonstrating that the mixing efficiency of the Barrier EWOD chip is six times higher than that of the Re EWOD Chip. Additionally, it is worth noting that the Re EWOD Chip required more mixing to achieve a complete reaction, further highlighting the effectiveness of the Barrier EWOD Chip.

## 4. Conclusions

Given the significant challenges associated with mixing large molecules on digital microfluidic platforms, along with the practical requirement for maximum area utilization on the chip, we propose an electrode design featuring a micro-barrier for simultaneous mixing during droplet transport. This approach requires no additional equipment or area, while the asymmetric design further enhances the chip’s area utilization. The asymmetric design enables the driving and mixing of multiple droplets with an array of micro-barrier electrodes using only two signals. The study validated and analyzed the newly designed fast mixing capability through experiments and simulations and successfully applied it to the detection process of sorbitol. This approach provides a simple and fast solution to the mixing problem of large molecules on a chip.

## Figures and Tables

**Figure 1 sensors-23-07102-f001:**
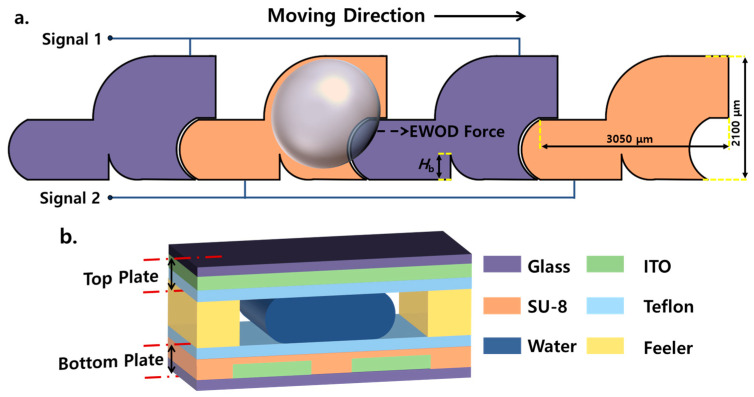
Schematic diagram of EWOD equipment. (**a**) Design of electrodes with a micro-barrier. The height of the micro-barrier is *H*_b_. The electrode array is connected to two opposite electrical signals, with the active electrode filled with purple and the opposite filled with orange. The electrodes are alternately activated, and the droplet moves to the right. Detailed design drawings are in Appendix A. (**b**) A side view of the structure of the EWOD device.

**Figure 2 sensors-23-07102-f002:**
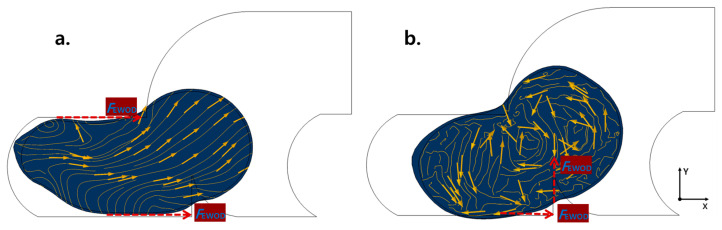
Internal streamline of the droplets. (**a**) At the end of step 1, the horizontal *F*_EWOD_ is still dominant and the streamline is mostly horizontal. At this point, the droplet starts to pass through the Barrier, and the tiny amount of *F*_EWOD_ makes the internal streamline start to bend slightly. (**b**) The droplet completely passes through the Barrier at this time, the horizontal and vertical *F*_EWOD_ act together, and vortices are formed inside the droplet.

**Figure 3 sensors-23-07102-f003:**
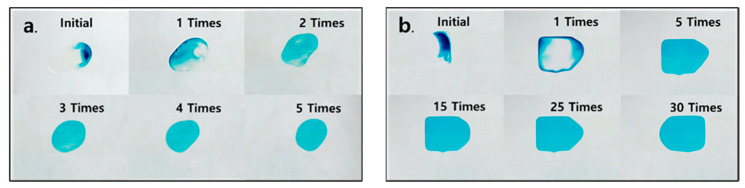
The distribution of the dye in the droplets varies with the number of mixings or time elapsed. The initial volume of all droplets is 0.65 μL, complete process is shown in Appendix A. (**a**) The blue dye in DI water with Barrier EWOD Chip. (**b**) The blue dye in DI water with Re EWOD Chip. (**c**) Blue dye diffusion in DI water.

**Figure 4 sensors-23-07102-f004:**
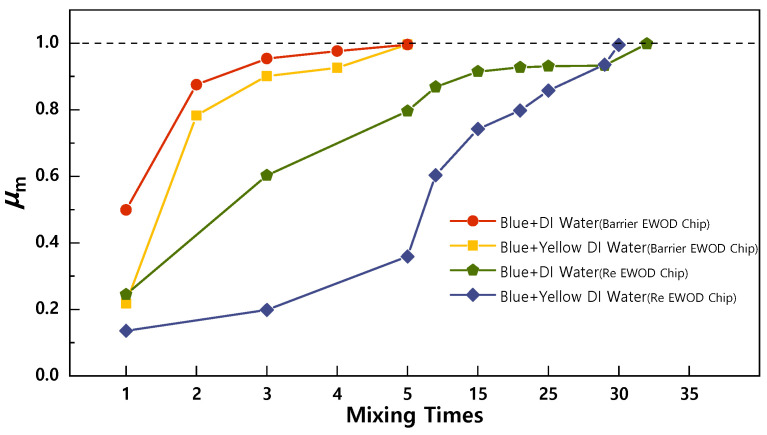
Mixing degree at different times of mixing.

**Figure 5 sensors-23-07102-f005:**
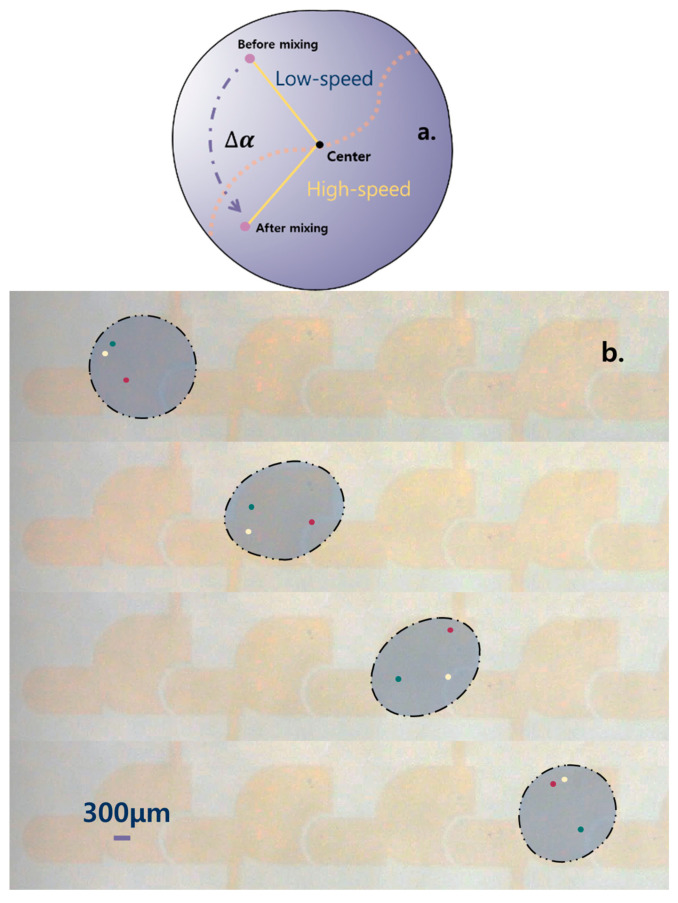
Partitioning and particle motion within a droplet. (**a**) The method of defining Δ*α*, high and low speed zones. (**b**) The particles within the droplet rotate, and the diameter of the particles is 80 μm and *H*_b_ is 200 μm. The three colors represent three particles.

**Figure 6 sensors-23-07102-f006:**
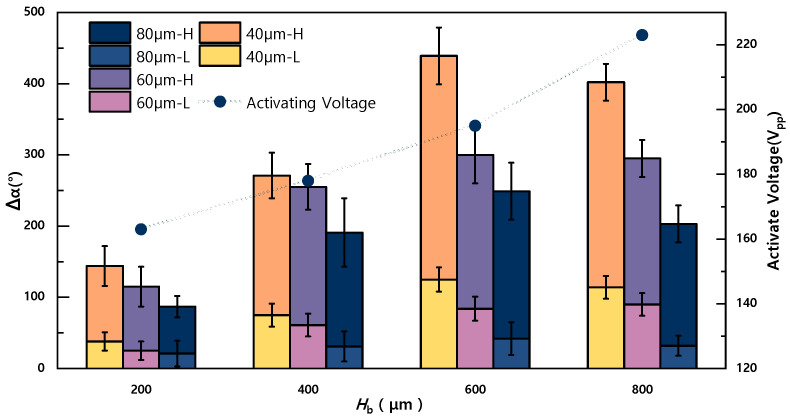
The effect of the barrier critical size *H*_b_ on Δ*α*. Where L means a low-speed region and H means a high-speed region.

**Figure 7 sensors-23-07102-f007:**
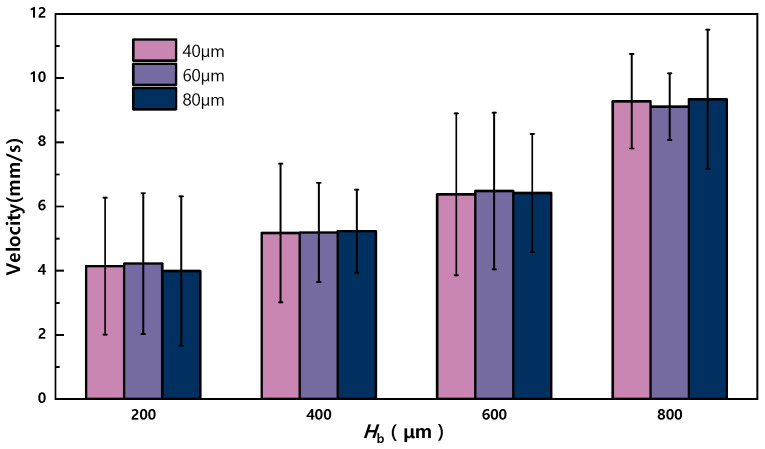
Effect of different particle sizes and different *H*_b_ values on activation velocity.

**Figure 8 sensors-23-07102-f008:**
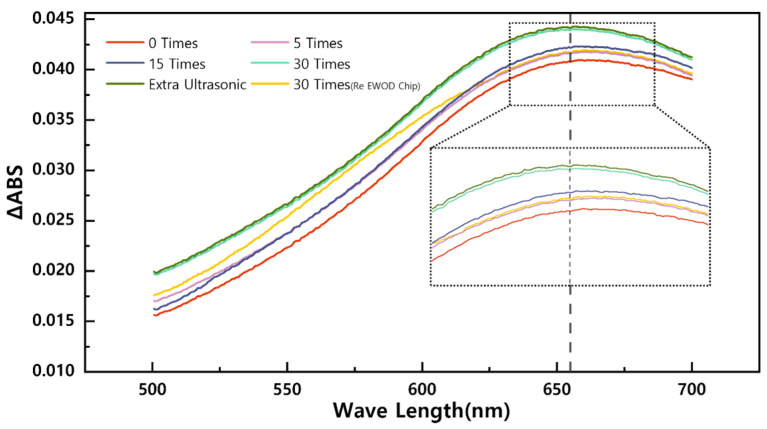
Detection results of different times of mixing and comparison groups.

## Data Availability

The data that support the findings of this study are available from the corresponding author upon reasonable request.

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
