# Peer review of "EWOD Chip with Micro-Barrier Electrode for Simultaneous Enhanced Mixing during Transportation"

_sensors, 2023, doi:10.3390/s23167102_

Round 1

Reviewer 1 Report

The manuscript title ‘EWOD Chip with Micro-Barrier Electrode for Simultaneous Enhanced Mixing during Transportation’ is an interesting work in the field of microfluidics.

I have the following queries regarding the manuscript.

1.      Please explain the term ‘Enhanced’ in the title of manuscript.

2.      The scheme is not clear. Please redraw the scheme for better clarity and add a good quality image of EWOD chip.

3.      Please provide more details in the figure 2 as described in the text. Please redraw the figure.

4.      How did authors measure 0.65 μL of sample volume. Why chose 0.65 μL not 1 μL for better calibration?

5.      Please carry out the experiments of particle size below 40 μm in diameter (e.g. 20, 10 μm) for a broader understanding of the particle size effect on mixing of liquids?

Reviewer 2 Report

The authors have presented a novel approach integrating micro-barriers on EWOD electrodes by optimizing the geometry of the transport electrode. The mixing performance was evaluated through a series of tests. The ideas are interesting and the experimental approach is sound. The paper should be interesting to the EWOD community.

Here are my suggestions:

1.   Introduction: since optimizing the electrode geometry is critical in this work, it will be nice to include a general survey about how EWOD researchers lately have tried to changing the electrode geometry to bring better functionality to the EWOD devices. A few more references will be encouraged.

2.  One of the advantages in EWOD chip is its programmability. In other words, the droplets can move in different routing on the electrodes when needed. Can authors comment (or have a paragraph in Discussion) on how the droplets behave when it moves to LEFT, instead of RIGHT, in Figure 1.

3.  Is each data point in Figure 4 coming from one droplet?

4.  It is difficult to recognize the geometry of the electrode and its relationship to the droplet in Figure 5b. Please improve Figure 5b.

5.  The authors may want to elaborate the preparation and material properties of Sorbitol. Particularly whether it may have different material (or solution) properties affect the EWOD motion, or its behavior on Barrier electrode?

6.  Figure 8 has the y-axis in (delta_ABS), instead of relative absorption. Can authors elaborate the implication of Figure 8?

7.  The authors are also encouraged to include a few related references from MDPI

Round 2

Reviewer 1 Report

Authors have addressed all the queries. Figures in the mansucript looks better than previous one. The manuscript can be accepted for publication.